

# Nonlinear forcing mechanisms of the terdiurnal solar tide and their impact on the zonal mean circulation

Friederike Lilienthal[1] and Christoph Jacobi[1]

[1]Institute for Meteorology, Universität Leipzig, Stephanstr. 3, 04103 Leipzig, Germany

**Correspondence:** F. Lilienthal (friederike.lilienthal@uni-leipzig.de)

**Abstract.** We investigate the forcing mechanisms of the terdiurnal solar tide in the middle atmosphere using a mechanistic global circulation model. In order to quantify their individual contributions, we perform several model experiments and separate each forcing mechanism by switching off the remaining sources. We find that the primary excitation is owing to the terdiurnal component of solar radiation absorption in the troposphere and stratosphere. Secondar y sources are nonlinear tide-tide interactions and gravity wave-tide interactions. Thus, although the solar heating clearly dominates the terdiurnal forcing in our simulations, we find that nonlinear tidal and gravity wave interactions contribute in certain seasons and altitudes. By slightly enhancing the different excitation sources, we test the sensitivity of the background circulation on these changes of the dynamics. As a result, the increase of terdiurnal gravity wave drag can strongly affect the middle and upper atmosphere dynamics, including an irregular change of the terdiurnal amplitude, a weakening of neutral winds in the thermosphere, and a significant temperature change in the thermosphere, depending on the strength of the forcing. On the contrary, the influence of nonlinear tidal interactions on the middle atmosphere background dynamics is rather small.

## 1 Introduction

The middle atmosphere dynamics are mainly determined by waves that are excited in the troposphere or stratosphere and propagate to the upper atmosphere (see, e.g., reviews by Forbes, 1995; Yiğit and Medvedev, 2015). These waves can be either global scale like atmospheric solar tides, or small scale like the internal gravity waves (GW). GWs are generated in the lower atmosphere due to orography, convective instabilities, wind shears, jet streams, spontaneous adjustment, or wave–wave interactions (Fritts and Alexander, 2003). Due to wave breaking and momentum deposition they are mainly responsible for the wind reversal in the mesosphere and lower thermosphere (MLT) region. However, GWs also play an important role in the thermosphere where they can damp or enhance tides (e.g., Yiğit et al., 2008; Yiğit and Medvedev, 2017), and may also transport wave signatures to the thermosphere (e.g., Eckermann et al., 1997; Meyer, 1999; Hoffmann et al., 2012).

Atmospheric solar tides are global scale waves owing to the diurnal variation of solar radiation. Therefore, they have periods of a solar day and its harmonics. They are primarily excited in the water vapor and ozone heating region (Chapman and Lindzen, 1970; Andrews et al., 1987). Due to decreasing density with increasing height, tides reveal their maximum amplitudes in the MLT region. Above, in the thermosphere, they are damped, e.g. by increasing molecular diffusion and thermal conduction.



Tides modulate the background wind field and therefore have an impact on the propagation conditions of GWs (e.g., Eckermann and Marks, 1996; Senf and Achatz, 2011; Yiğit and Medvedev, 2017; Baumgarten et al., 2018).

Amplitudes of diurnal tides (DTs) and semidiurnal tides (SDTs) are generally larger than those related to higher harmonics and wavenumbers such as the terdiurnal tide (TDT). However, during some seasons the TDT amplitudes may locally become

comparable to those of the DT (Cevolani and Bonelli, 1985; Reddi et al., 1993; Thayaparan, 1997; Younger et al., 2002). For example, radar measurements at midlatitudes show large TDT amplitudes in autumn and early winter (Beldon et al., 2006; Jacobi and Fytterer, 2012; Jacobi, 2012), and also in spring (Thayaparan, 1997). Global observations of the TDT have been presented by Smith (2000), Moudden and Forbes (2013), Pancheva et al. (2013) and Yue et al. (2013). Yue et al. (2013) obtained TDT amplitudes of more than $16\,\mathrm{m\,s^{-1}}$ at $50°$ N/S above $100\,\mathrm{km}$ from observations using the Thermosphere Ionosphere Mesosphere

Energetics and Dynamics Doppler Interferometer (TIMED/TIDI). They reported another maximum in the meridional wind at about $82\,\mathrm{km}$ at lower northern latitudes. For $90\,\mathrm{km}$ altitude, based on Sounding of the Atmosphere using Broadband Emission Radiometry (SABER) data, Moudden and Forbes (2013) observed large amplitudes over the equator during equinoxes $(6 - 8\,\mathrm{K})$, and also at $60°$ during spring.

While the excitation mechanism is relatively well known for the DT and SDT, those of the TDT are still under debate

(e.g., Lilienthal et al., 2018, and references therein). Besides the direct solar forcing, higher harmonics are also subject of nonlinear tidal interaction (e.g., Glass and Fellous, 1975; Teitelbaum et al., 1989; Teitelbaum and Vial, 1991). For example, the interaction between DT and SDT can lead to a secondary TDT. On the other hand, interactions between GWs and tides can also produce a secondary TDT (Miyahara and Forbes, 1991). Ribstein and Achatz (2016) have shown that such interactions strongly depend on model physics but they did not include the TDT in their analysis.

The excitation mechanisms of the TDT have been investigated by several model studies (Akmaev, 2001; Smith and Ortland, 2001; Huang et al., 2007; Du and Ward, 2010; Lilienthal et al., 2018) but with partly inconclusive results. This is most likely caused by different models and analysis techniques, e.g. Akmaev (2001) and Smith and Ortland (2001) use models with explicit lower boundary forcing of DT and SDT while the simulations of Du and Ward (2010) and Lilienthal et al. (2018) are based on fully self-consistent tides. Furthermore, the authors partly focused on different latitudes and altitudes which cannot be easily

compared.

The majority of these publications agrees that the direct solar forcing is the most dominant, although not the only, excitation mechanism of the TDT (Akmaev, 2001; Smith and Ortland, 2001; Du and Ward, 2010; Lilienthal et al., 2018). For example, Huang et al. (2007) found significant nonlinear TDT amplitudes above $90\,\mathrm{km}$, especially during equinoxes, which agrees with the simulations by Akmaev (2001). Lilienthal et al. (2018) found that the solar forcing is the primary excitation mechanism,

but nonlinear tide-tide interactions and also GW-tide interactions play a role. They analyzed the phase relations of differently forced TDTs and found destructive interferences between them. This suggests that different excitation mechanisms can also counteract, partly leading to a reduced and not an enhanced TDT.

To extend the work of Lilienthal et al. (2018) and in order to further investigate the nonlinear mechanisms of TDT forcing, we now present model simulations, which are each restricted to only one terdiurnal forcing mechanism, i.e. either the solar

heating absorption, or nonlinear tidal interactions, or GW-tide interactions. The remainder of this paper is organized as follows.





In section 2, the model and experimental setup is described, and in section 3 the results of the model runs are discussed with respect to TDT zonal wind amplitudes (section 3.1). In section 3.2 a sensitivity study with modified forcings is presented and their effect on the mean flow is analyzed. Section 4 concludes the paper.

## 2   Model Description and Experimental Setup

In the following analysis we use the Middle and Upper Atmosphere Model (MUAM) in the same configuration as described in detail by Lilienthal et al. (2017, 2018). In short, MUAM is a mechanistic primitive equation global circulation model that reaches from the troposphere to the thermosphere, i.e. to about $160\,\mathrm{km}$ in logarithmic pressure height, given a constant scale height of $7\,\mathrm{km}$. The horizontal resolution is $5 \times 5.625°$ in latitude and longitude. The model's zonal mean temperature in the troposphere and lower stratosphere is nudged by monthly mean zonal mean ERA-Interim reanalysis (ERA-Interim, 2018; Dee

et al., 2011) temperatures. To provide ensemble simulations for each of the following experiments, 11 ensemble members are driven by monthly mean ERA-Interim reanalysis of the years 2000 to 2010. In this way, a set of ensembles also represents some kind of interannual variability, as shown by Lilienthal et al. (2018).

There are three main sources of atmospheric tides in the model. The primary source is the absorption of solar radiation which creates tides self-consistently. The solar heating is parameterized according to Strobel (1978) and considers heating due to all

important gases for tidal forcing such as water vapor and ozone in the troposphere and stratosphere, as well as oxygen and nitrogen in the thermosphere. For more details, see Lilienthal et al. (2018).

Nonlinear interactions between different tides and between GWs and tides can generate a secondary TDT as described above. The interactions related to GWs can be realized within the GW parameterization of the model. This is a coupled parameterization based on an updated linear scheme for the lower and middle atmosphere (Lindzen, 1981; Jakobs et al.,

1986; Fröhlich et al., 2003; Jacobi et al., 2006) and an adjusted nonlinear scheme according to Yiğit et al. (2008, 2009), see also Lilienthal et al. (2017, 2018), for the thermosphere. Even though both schemes are coupled through the eddy diffusion coefficient, which is transfered from the linear scheme to the nonlinear scheme, both parameterizations are almost independent from each other, because they handle a different range of phase speeds without overlap, i.e. the linear scheme is responsible for slowly traveling GWs which mainly break in the middle atmosphere while the nonlinear scheme is responsible for fast

traveling GWs that reach the lower thermosphere.

Nonlinear interactions are a rather dynamic feature of the tendency equations of the model. They are, mathematically, to a certain degree hidden in the product of non-zonal parameters of the model equations. In particular, they are included in the advection terms and in the adiabatic heating component (see also Lilienthal et al., 2018). Further interactions, i.e. further products of non-zonal parameters, are possible within the parameterizations of eddy diffusion, molecular conduction, and in

the Coriolis terms. However, these terms are comparatively small and their separation and extraction, partly leads to numerical instabilities. Therefore, these terms are neglected in the following. To summarize, the main three forcing mechanisms of TDTs in MUAM are the direct solar forcing, nonlinear tidal interactions and GW-tide interactions.



In order to quantify the relevance of these three mechanisms, Lilienthal et al. (2018) performed model runs, each of them with removing one of these mechanisms in order to determine the change in tidal amplitude due to this forcing. Following Lilienthal et al. (2018) and extending the analysis, we now go the other way around and remove all forcing mechanisms except for the respective one of interest. The procedure to remove the nonlinear terms is technically the same as the one used by Lilienthal et al. (2018). A Fast Fourier transform according to Danielson and Lanczos (1942) is used to extract the wavenumber 3 pattern in each time step of the model. Due to the fact that the model, in the current configuration, does not generate nonmigrating tides, this is the simplest way to remove the whole TDT structure. In contrast to Lilienthal et al. (2018), this is not only applied to one of the forcing terms, but to two of them in parallel. The remaining amplitudes can be directly attributed to the respective third and remaining forcing. Thereby we produce a reference simulation with all forcing mechanisms included, and three further simulations:

- `REF`: reference run. This is the same one that has been shown by (Lilienthal et al., 2018),

- `SOL`: no nonlinear and GW forcing; TDT amplitudes are only owing to the absorption of solar radiation,

- `NLIN`: no solar and GW forcing; TDT amplitude are only owing to nonlinear tidal interactions,

- `GWF`: no solar and nonlinear forcing; TDT amplitudes are only owing to GW-tide interactions.

All of these simulations are performed as ensembles as described above.

In order to investigate the impact of these forcing mechanisms on the background circulation, we also enhance the respective remaining forcing in the simulations `SOL`, `NLIN` and `GWF`, stepwise. Thereby, each simulation represents a certain factor of enhancement. Technically, this is the same procedure like that one for the removal of terdiurnal forcing terms, except that the respective wavenumber 3 forcing is increased by a certain factor. In order to reduce the time of computation for the simulations with enhanced forcing mechanisms, only the January runs are performed as an ensemble. The other months represent the conditions for the year 2000, only.

## 3 Results

### 3.1 Zonal wind amplitude distributions for different forcing mechanisms

Fig. 1 shows, for the four simulations, the seasonal cycle of zonal wind amplitudes at $109\,\mathrm{km}$, averaged over the 11 ensemble members. The `rREF` results are shown in Fig. 1a. Amplitudes at midlatitudes maximize in winter, and secondary maximums are seen during the equinoxes. Radar measurements at northern midlatitudes also show this seasonal cycle (e.g., Teitelbaum et al., 1989; Thayaparan, 1997; Jacobi and Fytterer, 2012; Jacobi, 2012; Fytterer et al., 2013; Yu et al., 2015) and satellite measurements by Yue et al. (2013) agree in the southern hemisphere. Above the equator, amplitudes maximize during equinoxes and at low latitudes during local summer which is in accordance with, e.g., Deepa et al. (2006), Venkateswara Rao et al. (2011), Guharay et al. (2013), or Yu et al. (2015). Poleward of $70°$, the amplitudes of the TDT rapidly decrease. Note that the

©c Author(s) 2019. CC BY 4.0 License.




amplitudes of the REF simulation are generally smaller than the observed ones. This is due to the fact that MUAM tends to underestimate tides in general, which is frequently seen in other models, too (Smith, 2012; Pokhotelov et al., 2018; Lilienthal et al., 2018).

It is obvious that the SOL simulation (Fig. 1b) has TDT amplitudes similar to the REF ones, again with maxima in local winter of the midlatitudes. Visually the differences to the REF run are small. This already shows that solar heating is the major source of the TDT in the middle atmosphere. However, there are some clear differences. So the midlatitude winter amplitudes of the SOL simulation are larger than the REF ones, while the equinox amplitudes are reduced both at middle and equatorial latitudes. As demonstrated by Lilienthal et al. (2018), the larger SOL amplitudes are mainly owing to destructive phase relations between the propagating TDTs excited by solar heating and nonlinear tidal interactions, respectively. A similar behavior has been reported by Smith et al. (2004) for the quarterdiurnal (6 h) tide.

The nonlinearly forced TDT (Fig. 1c) has a very similar seasonal cycle like the TDT that is directly excited by solar radiation, but its amplitudes are much smaller by a factor of 2 to 3. The nonlinear TDT has been modeled earlier (e.g., Smith and Ortland, 2001; Huang et al., 2007), but their seasonal cycle is considerably different from our model results, i.e. the earlier simulations led to maxima at low latitudes (Smith and Ortland, 2001) and during equinoxes (Huang et al., 2007).

On an average, the amplitudes of the GWF simulation (Fig. 1d) are smaller than those of the NLIN simulation but they maximize during summer near 60°N where they can be even larger than those due to the nonlinear forcing. Near the equator, they are close to zero. When the GWF amplitudes maximize, they reach a similar magnitude like those obtained by Miyahara and Forbes (1991).

In the following, we present the vertical structure of the TDT for different forcing terms and each season (Fig. 2). Background colors denote ensemble means of the latitude-altitude distribution, according to the 11 ensemble members, and contour lines represent their standard deviations. January and July reveal a similar amplitude distribution, considering that the hemispheric structure is reversed due to the reversed circulation patterns. April and October are similar, too. Standard deviations are generally small with respect to the ensemble means, and they tend to be larger in July (October) compared to January (April). Furthermore, the latitudinal structure of TDT amplitudes changes in the lower thermosphere (approximately above 120 km) for most experiments and seasons, e.g. the triple peak structure of REF and SOL in the MLT during equinoxes turns into a double peak structure above 140 km. In the following analyses, we restrict ourselves to the months January and April to show typical solstice and equinox conditions, respectively.

## 3.2 Impact of different forcing mechanisms on tidal amplitudes and background

In this section we analyze the effect of each different forcing on the TDT as well as the background atmosphere. Therefore, the simulations SOL, NLIN and GWF now serve as a reference for the TDT amplitudes and the respective background circulation. In each of these simulations, we enhance the active forcing mechanism (tendency term) in each time step and for each latitude/altitude by 5% of the respective original value, i.e. the solar forcing is enhanced in SOL, the nonlinear forcing is enhanced in NLIN and the GW forcing is enhanced in GWF. These enhanced simulations are called SOL5, NL5 and GW5.





Figure 3 shows the observed amplitude change of the respective terdiurnal forcing terms for January (top) and April (bottom) in the thermal (a,b,e,f) and dynamical (c,d,g,h) parameters. Thereby, the data at each grid point are normalized by their value in the respective reference simulation. For example, the terdiurnal nonlinear forcing of NL5 is normalized by the terdiurnal nonlinear forcing of the NLIN simulation. The solar forcing term of the SOL5 simulation is not shown in Fig. 3 because the

effect is nearly linear, i.e. the strength of enhancement almost shows the expected value of $+5\%$ with a maximum deviation between $+4.6\%$ to $+5.3\%$ during solstices. Figure 3 demonstrates that the observed nonlinear (NL5) and GW tendency terms (GW5) can strongly deviate from $+5\%$ compared to NLIN and GWF, respectively. The nonlinear forcing terms (temperature advection and zonal wind acceleration; Fig. 3a,c,e,g) show a change in the terdiurnal forcing between roughly $-8\%$ to $+22\%$. However, as indicated by the shaded areas, these are rather exceptional cases, and usually the forcing enhancement varies

between 4.5 to 5.5%. The GW forcing terms are more extreme, ranging from $-92\%$ to $+500\%$ in the heating component (Fig. 3b,f) and from $-93\%$ to almost $+1800\%$ in the component of zonal wind drag (Fig. 3d,h). The shading for the enhanced GW forcing covers a range between 0 to $10\%$, but shaded areas are rather small, indicating that these large numbers are not only outliers.

    A possible reason for these large discrepancies are feedback mechanisms within the model. It is widely known (e.g., Lindzen,

1981; Holton, 1982) that GWs strongly influence the background circulation, being responsible for the wind reversal in the mesosphere due to wave breaking and momentum deposition. Therefore, a change in the terdiurnal component of GW drag may also influence the background circulation, leading to altered propagation condition for tides, which again affects the terdiurnal component of GW drag. Such a mechanism is very difficult to control within a nonlinear model. Before we go into detail with the analysis of the background circulation, we first have a look at the amplitude of the TDT due to the increased forcing.

    Fig. 4 shows the January mean latitude-altitude distribution of TDT zonal wind (a,c) and temperature (b,d) amplitudes of the simulations NL5 (a,b) and GW5 (c,d). The vertical profiles to the right of each latitude-altitude distribution show the monthly mean horizontal mean relative amplitude changes $\Delta A$ for NL5 and GW5 of the year 2000 where

$$\Delta A_{\mathrm{NL5}} = (A_{\mathrm{NL5}} - A_{\mathrm{NLIN}})/A_{\mathrm{NLIN}} \cdot 100$$

and

$$\Delta A_{\mathrm{GW5}} = (A_{\mathrm{GW5}} - A_{\mathrm{GWF}})/A_{\mathrm{GWF}} \cdot 100.$$

The simulation NL5 (Fig. 4a) looks equal to NLIN (Fig. 2 first row, third column), because the enhancement of $5\%$ is too small to be visible in the chosen color scheme. The variability between the different seasons (Fig. 4a, profiles) is small for all altitudes, and the zonal wind amplitudes in NL5 are approximately $4\%$ to $6\%$ larger than in NLIN. The temperature amplitude (Fig. 4b) shows a similar behavior below $130\,\mathrm{km}$ but above, the horizontal mean TDT amplitudes roughly vary between $0\%$ and $+8\%$ compared to the original forcing. During April (light blue line), the amplitude does even slightly decrease near $150\,\mathrm{km}$,

i.e. the change is negative.

    The zonal wind TDT amplitudes due to a $5\%$ increased GW forcing (Fig. 4c) are drastically increased in comparison to the GWF amplitudes in Fig. 2 (first row, last column). This is mainly an effect of the increased GW forcing terms. They do not only influence the zonal wind amplitude (Fig. 4c) but also the temperature tide (Fig. 4d). Below $100\,\mathrm{km}$, the amplitudes are





approximately doubled (+100%). This factor further increases up to an altitude of 140 km reaching a maximum increase by more than 800% (zonal wind) and almost 600% (temperature), respectively. These maxima are found during August/September for both parameters. The change in amplitude is enormous, considering that the GW forcing was only increased by 5% in each time step. However, the overall change in the forcing locally amounts to +500% (in the heating due to GWs, see Fig.3b) and to

almost +1800% (in the zonal GW drag, see Fig.3h), which is possibly due to feedback mechanisms within the model that also influence the background conditions and GW propagation conditions. Therefore, the dramatic increase in TDT amplitude can be partly explained by the strongly enhanced GW forcing. Furthermore, the TDT amplitude changes are considerably strong above 100 km, which coincides with the fact that the terdiurnal zonal GW drag maximizes in the thermosphere (Fig.3).

    The differences of the zonal mean zonal wind and zonal mean temperature are shown in Fig. 5, each for January and April

conditions. We only show the differences between GW5 and GWF, and not between NL5 and NLIN because the latter ones are small. It can be seen in Fig. 5 that the terdiurnal GW forcing only affects the thermosphere. Below 130 km, the thermosphere experiences a cooling and above that height, there is a warming (Fig. 5b,d). This respective cooling and warming is larger in April than in January. The zonal wind during January (Fig. 5a) is mainly accelerated in the eastward direction, with a maximum at low and middle latitudes of the NH above 130 km. As a result, the westerly winds in that region are slightly enhanced by

about $+4\,\mathrm{m\,s^{-1}}$. During April, the zonal wind in the thermosphere is generally small. The enhanced GW forcing leads to an alternating pattern of eastward and westward directed acceleration with a magnitude of about $\pm 5\,\mathrm{m\,s^{-1}}$, again with maxima in the NH. The magnitude of the cooling/warming of the thermosphere strongly depends on the strength of the terdiurnal GW forcing, i.e. it becomes stronger for stronger enhancements (not shown here).

    Figure 6 shows the behavior of TDT amplitudes depending on different factors of forcing enhancements where a factor

of 1.05 refers to an increase by 5%. Figure 6a,b refer to an increase of the terdiurnal nonlinear forcing in steps of 10% enhancement, and Fig. 6c,d refer to an increase of the terdiurnal GW forcing in steps of 1%. The amplitude response is shown for the temperature (Fig. 6, top row) and the zonal wind component (Fig. 6, bottom row). The different colors refer to different months of the year 2000 and the relative amplitude change is globally averaged for a height range of 80 to 160 km.

    For the increased nonlinear forcing (Fig. 6a,b), a linear fit is added where each fit has a squared correlation coefficient

$R^2 > 0.99$ and the respective slopes are given in the legend. They are close to 1, suggesting that the amplitude is directly correlated with the factor of increase in the nonlinear forcing. However, this does not mean that the total observed amplitude in the REF simulation is increased by the same factor. The increase in amplitude only refers to the pure nonlinear part of the TDT. Due to the fact that the nonlinear TDT is much weaker than the solar TDT, its overall impact is rather small.

    The dependence of TDT amplitudes on the GW forcing (Fig. 6c,d), are irregular for implemented enhancements larger than

5%. For an increase of 10%, the model becomes instable for some months. This is certainly related to the influence of GWs on the zonal mean circulation, as shown exemplary for January and April in Fig. 5, and for all months in Fig. 7. This figure is similar to Fig. 6, but instead of amplitudes we show the global mean (80 to 160 km) absolute value of zonal mean differences to NLIN or GWF, respectively. Instead of the slopes, we show the correlation coefficients for the linear fits in the legend of Fig. 7a,b. These are close to one for most of the months, except for June to August (for zonal mean temperature) and for June



to September (for zonal mean zonal wind). However, the overall impact of nonlinear forcing mechanisms on the background circulation is small as global mean differences amount to less than $0.5\,\dot{\mathrm{K}}$ and $0.5\,\mathrm{m\,s^{-1}}$, respectively.

Again, the response is much more relevant, when the GW forcing is increased (Fig. 7c,d). The temperature reveals an exponential-like increase in absolute temperature change where an increase by $10\%$ can change the zonal mean temperature above $80\,\mathrm{km}$ by more than $50\,\mathrm{K}$ on a global average and the zonal mean zonal wind by about $2$ to $6\,\mathrm{m\,s^{-1}}$. The maximum temperature change is found in the thermosphere as shown in Fig. 5. To give an example, an increase of the GW forcing by $10\%$ in January leads to a temperature decrease of about $10\,\mathrm{K}$ in the mesosphere (about $110\,\mathrm{km}$ altitude). The patterns of the differences are in this case similar to those shown in Fig. 5. In the thermosphere, the temperature is drastically increased by up to $140\,\mathrm{K}$ near the upper boundary of the model.

## 4 Conclusions

Based on the experiments by Lilienthal et al. (2018), we performed extended simulations of the terdiurnal solar tide using a mechanistic global circulation model. Besides the primary forcing, which is the absorption of solar radiation in the lower atmosphere (Chapman and Lindzen, 1970; Andrews et al., 1987), further possible sources of atmospheric tides are nonlinear tidal interactions (e.g., Glass and Fellous, 1975; Teitelbaum et al., 1989) and gravity wave-tide interactions (e.g., Miyahara and Forbes, 1991; Ribstein and Achatz, 2016).

In order to separate the forcing mechanisms, we performed simulations in which we kept only one of these forcings and removed the other sources. As a result, these simulations allowed us to show the amplitudes of the TDT based on each excitation mechanism, separately, and we found that the global structure of the simulated TDT (REF simulation) is in good accordance with measurements in the MLT. Furthermore, the pure solar forcing (SOL simulation) explains most of the TDT global structure. This, in combination with the small TDT amplitudes of NLIN and GWF, indicates that the direct solar heating is the most important excitation mechanism of the TDT. Nonlinear tidal interactions only play a role during local winter at midlatitudes above $100\,\mathrm{km}$ and during equinoxes above $140\,\mathrm{km}$. GW-tide interactions mainly appear in the thermosphere with maxima during NH summer and during equinoxes above the equator.

The influence of the nonlinear tidal and GW-tide interactions on TDT amplitudes and on the zonal mean circulation was investigated based on a sensitivity study with enhanced terdiurnal forcing terms. Each simulation represented a certain factor of increase and we focused on the $5\%$ increased simulation which was the best compromise between significant changes in the atmosphere and numerical stability of the simulations. Our main results are the following:

– There is a direct and linear relationship between the nonlinear tidal forcing and the TDT amplitudes, but its influence on the zonal mean circulation is small.

– The influence of GW-tide interactions is more irregular with respect to the TDT amplitude, indicating that GW can play an important role for TDT forcing when the conditions for GW-tide interactions are favorable, especially in the thermosphere (e.g. Yiğit et al., 2008). Lilienthal et al. (2018) have shown that terdiurnal zonal GW drag is large in the thermosphere and this may also cause the large TDT amplitudes.




– There is an exponential-like relationship between GW-tide interactions and the zonal mean circulation in the thermosphere, which is cooled below 130 km and heated above. This is even more pronounced in April compared to January. Zonal wind in the thermosphere is slightly increased in January and has a more complex pattern in April.

Note that an artificial enhancement of the terdiurnal GW drag releases more energy into the system, i.e. GW amplitudes are larger causing GWs to reach higher altitudes. In the thermosphere, they release their energy due to wave breaking and can thereby strongly influence the dynamics in this region.

To conclude, modifications of terdiurnal forcing mechanisms do not only have an effect on TDT amplitudes but they may also influence the background circulation, especially with respect to the terdiurnal GW drag. Since tidal forcing in a real atmosphere is not as regular as in our model, such interactions may play an important role for the vertical coupling of the atmosphere. Our simulations also demonstrate the importance of GW-tide interactions and their consideration in global circulation models.

*Code availability.* The MUAM model code can be obtained from the corresponding author on request.

*Author contributions.* F. Lilienthal designed and performed the MUAM model runs. F. Lilienthal together with C. Jacobi drafted the first version of the text and discussed the results.

*Competing interests.* C. Jacobi is one of the Editors-in-Chief of Annales Geophysicae.

*Acknowledgements.* The authors acknowledge support through the Deutsche Forschungsgemeinschaft (DFG) under grant JA 836/30-1. SPARC global ozone fields were provided by W.J. Randel (NCAR) through ftp://sparc-ftp1.ceda.ac.uk/sparc/ref_clim/randel/o3data/. Mauna Loa carbon dioxide mixing ratios were provided by NOAA through ftp://aftp.cmdl.noaa.gov/data/trace_gases/co2/flask/surface/. ERA-Interim data have been provided by ECMWF on http://apps.ecmwf.int/datasets/data/interim_full_moda/?levtype1/4pl. We further acknowledge support from the German Research Foundation (DFG) and Universität Leipzig within the program of Open Access Publishing.



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





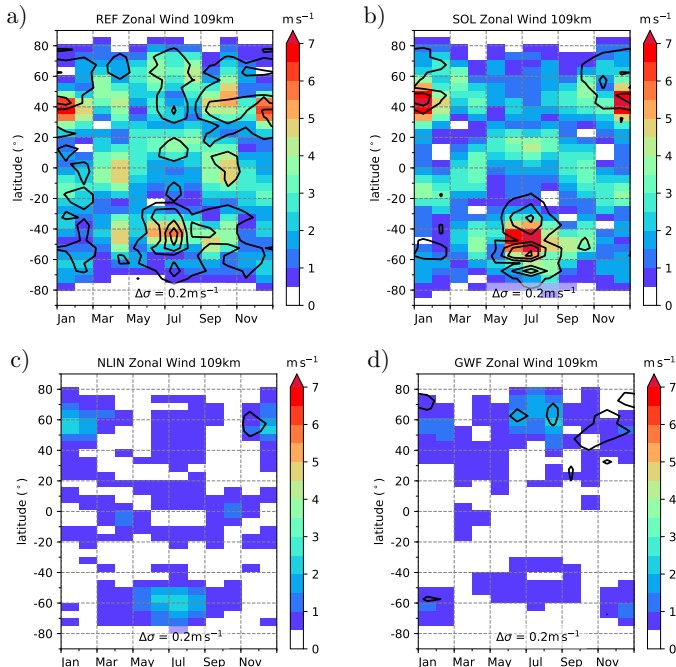

**Figure 1.** Seasonal cycle of TDT amplitudes at 109 km owing to different forcing mechanisms for the simulations a) `REF`, b) `SOL`, c) `NLIN` and d) `GWF`. Contour lines indicate standard deviations $\sigma$.





**Figure 2.** Latitude-altitude distribution of TDT amplitudes owing to different forcing mechanisms. From left to right: REF, SOL, NLIN, GWF. From top to bottom: January, April, July and October conditions. Contour lines indicate standard deviations $\sigma$.





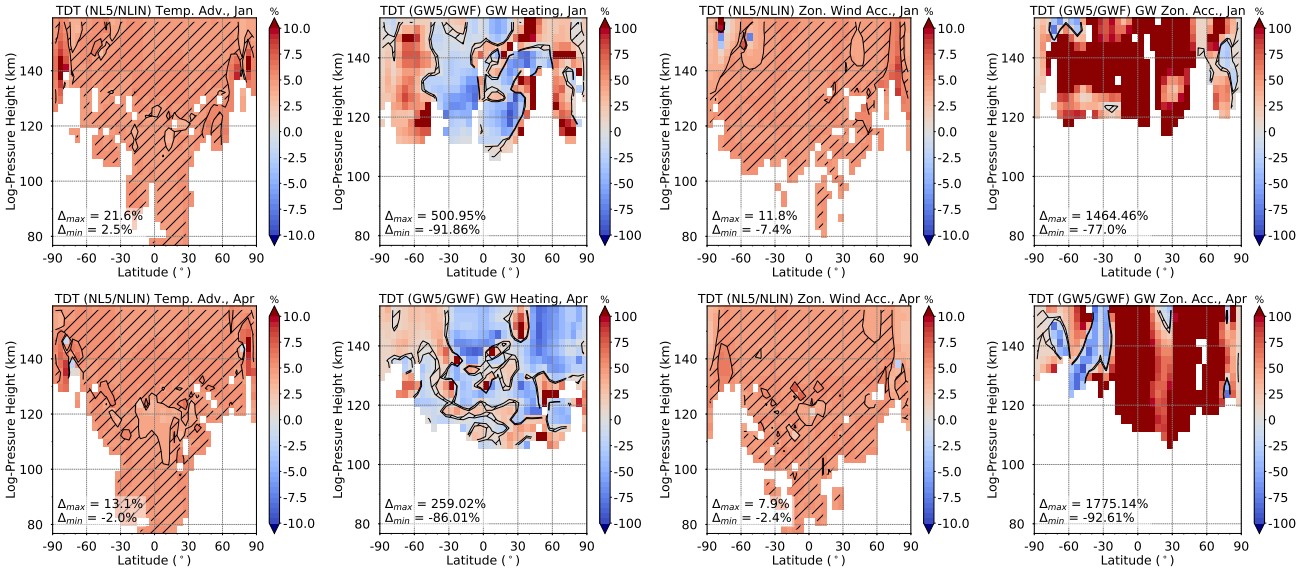

**Figure 3.** Relative change of terdiurnal forcing terms for an implemented increase of $5\%$ for July. From left to right: nonlinear temperature advection, heating due to GW-tide interactions, nonlinear zonal wind acceleration and zonal drag due to GW-tide interactions. Red (blue) colors refer to a larger forcing in NL5 and GW5 (NLIN and GWF). Hatched areas highlight the values $4.5\% \leq (\text{NL5}/\text{NLIN} - 1) \cdot 100 \leq 5.5\%$ and $0\% \leq (\text{GW5}/\text{GWF} - 1) \cdot 100 \leq 10\%$, respectively. Blank areas denote that the respective forcing in NLIN is smaller than $1\,\text{K}\,(2\,\text{m}\,\text{s}^{-1})$, and the respective forcing in GWF is smaller than $1\,\text{K}\,(10\,\text{m}\,\text{s}^{-1})$.





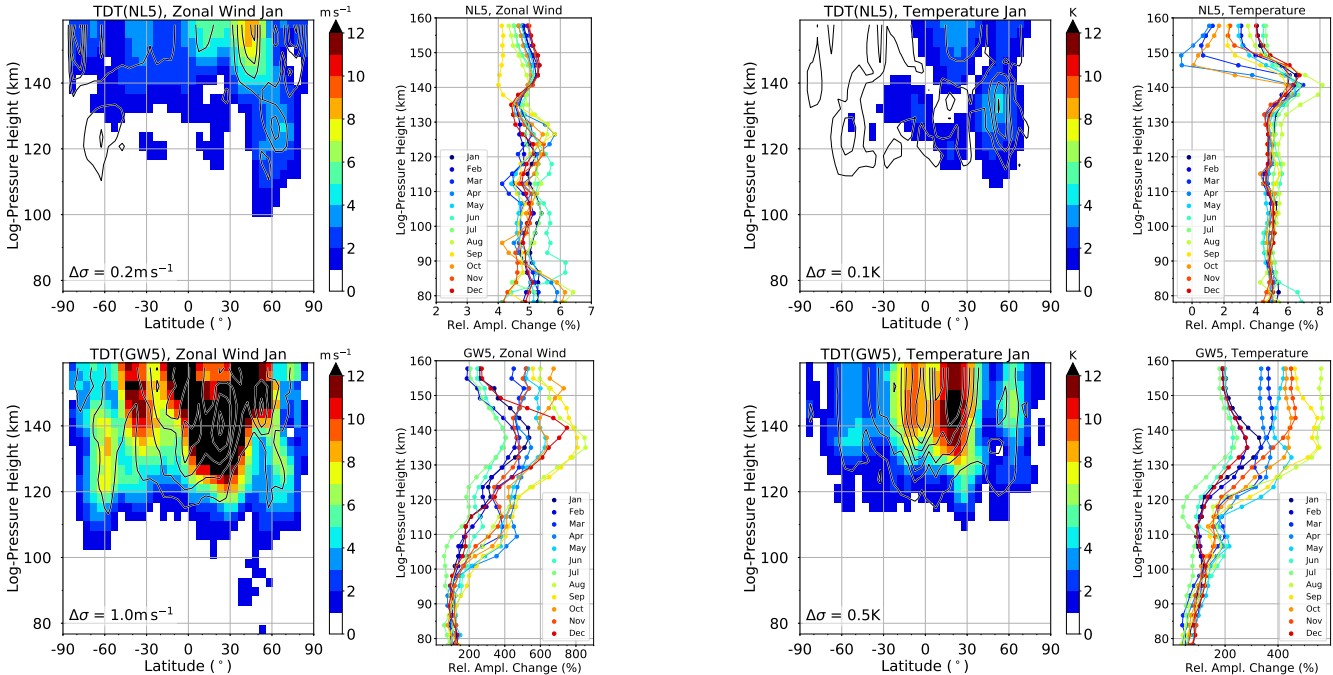

**Figure 4.** Color plots: Latitude-altitude distribution of (left) TDT zonal wind and (right) TDT temperature amplitudes in the simulations `NL5` (top row) and `GW5` (bottom row) for January 2000. Contour lines denote differences to the simulation `NLIN` (top) and `GWF` (bottom). Vertical profiles show the monthly mean horizontal mean TDT amplitudes, displayed as a relative change to `NLIN` or `GWF`, respectively. Different colors refer to different months (see legend). Note that scales are different.




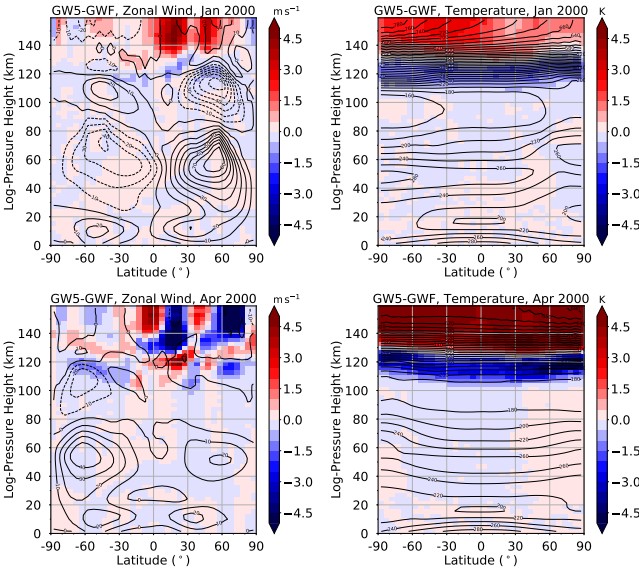

**Figure 5.** Contour lines: Latitude-altitude distribution of zonal mean zonal wind (left) and zonal mean temperature (right) in the simulation `GW5` for January (top) and April (bottom) 2000. Color shading denotes differences to `GWF`.

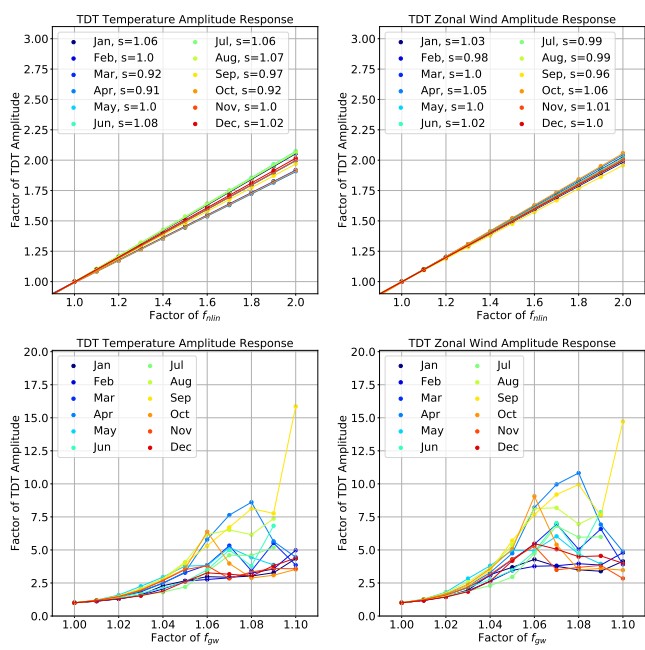

**Figure 6.** Normalized, vertically averaged (80-160 km height range) TDT amplitudes for temperature (left) and zonal wind (right) according to an increased terdiurnal forcing in `NLIN` (top) and `GWF` (bottom). Different colors refer to different months. Dots denote specific simulations. For `NLIN` (top), a linear fit is added for each month (corresponding slope $s$: see legend).





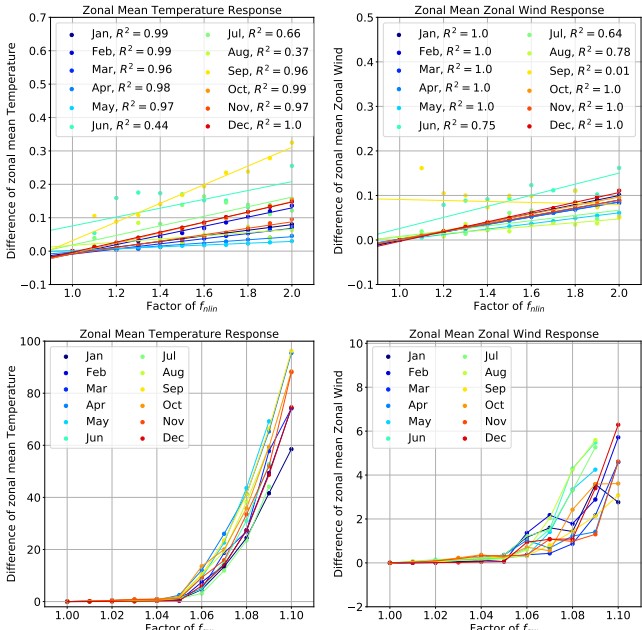

**Figure 7.** Global mean (80-160 km height range) absolute change of zonal mean temperature (left) and zonal wind (right) according to an increased terdiurnal forcing in `NLIN` (top) and `GWF` (bottom). Units are $m\,s^{-1}$ and $K$, respectively. For `NLIN` (top), a linear fit is added for each month (corresponding correlation coefficients $R^2$: see legend).