# Peer review of "Nonlinear forcing mechanisms of the terdiurnal solar tide and their impact on the zonal mean circulation"

_Annales Geophysicae, 2019_

## Referee Comment (RC1) · Anonymous Referee #1 · 14 Jun 2019

This paper investigates the forcing mechanisms of the terdiurnal solar tide in the middle atmosphere using a mechanistic global circulation model. Three are identified: Solar heating which is the most prominent, wave-wave interaction (between diurnal and semidiurnal tides) and tide-gravity wave interaction that are secondary. The authors also test the sensitivity of the background winds and temperatures to enhancements of the different sources. Incremental gravity wave drag is found to generate the strongest terdiurnal amplitude responses, and changes to the background winds.

The paper can be published, with the following revisions: The material reported in sections 1, 2, and 3.1 reprises findings reported in Lilienthal et al., 2018, and I saw

little value in it. I recommend that section 3.1 be cut, or very briefly summarized. The paper should only report the new information appearing in section 3.2 and beyond, and Figures 3 and onward.

I also recommend that it be edited for proper English grammar and usage.

Minor comments:

1. The text refers to panels a and b in Figure 4, but these labels that do not appear in the plots. Please recheck all Figures for this issue.

---

## Referee Comment (RC2) · Anonymous Referee #2 · 17 Jun 2019

Paper summary:

Base on the Middle and Upper Atmosphere Model (MUAM), this paper investigates the exact contribution of the forcing mechanisms of the terdiurnal solar tide (the absorption of solar radiation and the nonlinear interactions between different tides and between GWs and tides). The impact of these forcing mechanisms, especially the nonlinear interactions, on the terminal tidal amplitude and zonal mean circulation is also studied.

I have no major issues with this paper.

I would like the authors to discuss the following minor comments:

[Figure]

1. The model does not generate non-migrating tides in the current configuration as described in section 2. Please add "migrating" before "terdiurnal solar tide" in the title.

2. "In order to reduce the time of computation for the simulations with enhanced forcing mechanisms, only the January runs are performed as an ensemble. The other months represent the conditions for the year 2000, only". Please discuss the potential difference or evaluate the influence of these two kinds of precessing methods.

3. Figure 1 and Page 5 Line 1: "This is due to the fact that MUAM tends to underestimate tides in general, which is frequently seen in other models, too".

Page 5 Line 14: There are differences in the seasonal variation of the TDT from different models.

What's the reason behind these phenomena?

Besides, would you please demonstrate why you present the seasonal cycle of TDT at 109 km?

4. Figure 3 and Page 7 Line 4: ". . . in the forcing locally amounts to +500% . . . and to . . . +1800% . . .". The +500% and +1800% cannot be tell in Figure 3b and Figure 3h, although you have demonstrated the maximum and minimum values. You can add some red contours with the exact contour values in Figure 3 especially in Figure 3 (b, d, f, and h).

5. You can simply illustrate the limit of the standard deviations. Only the interval of the standard deviations in Figures 1, 2, and 4 can not demonstrate the exact values.

6. Page 7 Line 30: ". . .becomes instable for some months." The results of which months in Figure 6 are unstable?

Problems in figures and grammatical suggestions:

Figure 3:

Figure description and titles of the Figure 3 are inconsistent.

For example:

1. "July" in the figure description; "Jan and Apr" for the titles of Figure 3

2. "Zonal drag" due to GW-tide interactions in the figure description, "Zon. Acc." in the titles of Figure 3

3. a∼h is not noted in Figure 3. There are similar problems in Figures 4, 5, 6.

Figures 6 and 7:

1. What's the difference of the vertical mean and global mean? The results of which latitude are demonstrated in Figures 6 and 7.

2. You can add some descriptions about the "factor of fgw" in the manuscript.

Page 1 Line 4: "Scondar y sources" -> "Scondary sources"

Page 1 Line 15: "the internal gravity waves (GW)" -> "the internal gravity waves (GWs)"

Page 1 Line 16: "orography"->"geography"

Page 2 Line 15: "be subject of" -> "be subject to"?

Page 3 Line 18: "as described above" -> "as described in section 1"

Page 4 Line 25 "rREF" -> "REF"; "maximums" -> "maxima"

Page 5 Line 6: You presented the results at 109 km which can not represent the whole middle atmosphere. Thus, "in the middle atmosphere" is not exact in this sentence.

Page 5 Line 6: "So the midlatitude" -> "For example, the midlatitude"

Page 7 Line 25: "the respective slopes are given in the legend." -> "the respective slopes (s) are given in the legend."

---

## Author Comment (AC1) · 14 Jul 2019

Dear anonymous referee,

thank you very much for your valuable comments to help improve our manuscript. Please find below our detailed response to each of your concerns:

"The paper can be published, with the following revisions: The material reported in sections 1, 2, and 3.1 reprises findings reported in Lilienthal et al., 2018, and I saw little value in it. I recommend that section 3.1 be cut, or very briefly summarized. The paper should only report the new information appearing in section 3.2 and beyond, and

[Figure]

Figures 3 and onward."

We agree that sections 1, 2 and 3.1 present some similarities with Lilienthal et al. (2018). However, in contrast to the present paper, there we presented simulations with one of the forcing mechanisms removed and therefore, two of three mechanisms were still included. The new aspect of Figs. 1 and 2 is to present the absolute contribution of each single forcing. We will remove Fig. 1 and reduce Fig. 2, restricting ourselves to January conditions. Accordingly, the text in section 1-3.1 will be adjusted and shortened.

"I also recommend that it be edited for proper English grammar and usage."

We carefully went through the paper and corrected to our best knowledge, also considering the suggestions of reviewer #2.

"Minor comments: 1. The text refers to panels a and b in Figure 4, but these labels that do not appear in the plots. Please recheck all Figures for this issue."

Thank you very much for the hint. We double checked all figures and included the labels in each of them.

---

## Author Comment (AC2) · 14 Jul 2019

Dear anonymous reviewer,

thank you very much for your valuable comments to help improve our manuscript. Please find below our detailed response to each of your concerns:

"1. The model does not generate non-migrating tides in the current configuration as described in section 2. Please add "migrating" before "terdiurnal solar tide" in the title."

We adjusted the title, accordingly.

"2. "In order to reduce the time of computation for the simulations with enhanced

forcing mechanisms, only the January runs are performed as an ensemble. The other months represent the conditions for the year 2000, only". Please discuss the potential difference or evaluate the influence of these two kinds of precessing methods."

In the meantime, we can provide an update with ensemble means for the 5% enhanced simulations (GW5/NL5), too. This should be more consistent than using only one single year. Figure 4 will be updated in the revised version. Please note that Fig. 2 of the manuscript also includes a standard deviation with respect to the 11 ensemble members so that the reader may also get an idea of the year to year variability from these figures.

"3. Figure 1 and Page 5 Line 1: "This is due to the fact that MUAM tends to under-estimate tides in general, which is frequently seen in other models, too". Page 5 Line 14: There are differences in the seasonal variation of the TDT from different models. What's the reason behind these phenomena? Besides, would you please demonstrate why you present the seasonal cycle of TDT at 109 km?"

There is no general agreement on the cause of tidal underestimation in models, but it has to be connected with strong damping in the mesosphere region, for example due to gravity waves. The Lindzen-type parameterization used in MUAM tends to damp tides relatively strongly, while other parameterizations can lead to slightly stronger tides (e.g. the one after Hines or Yigit). Neverless, as we mainly focus on relative amplitude changes, our results are not significantly influenced by this underestimation.

Please note that our statement about the different seasonal cycles (page 5 line 14) only refers to the pure nonlinear TDT, not to the total TDT (which could be compared to observations). The reason is most likely based on the different tidal forcing mechanisms, but also on different methods to determine the nonlinear TDT contribution. For example, Smith and Ortland (2001) used a model with explicit lower boundary forcing of the diurnal (DT) and semidiurnal tides (SDT), while all tides in MUAM are forced in-situ by absorption of solar radiation without lower boundary forcing. The latitudinal

distribution of the tides by Smith and Ortland (2001) is therefore already prescribed to a certain degree and possible nonlinear interactions may appear at different latitudes and altitudes than in our model.

For Fig. 1, the altitude of 109km was chosen to be able to compare the REF results with satellite measurements which are not available further above. Below that altitude, MUAM produces relatively small amplitudes which might be connected with large uncertainties. Therefore, we think that 109km is the best choice as a compromise between large amplitudes and comparability. Note, however, that according to the suggestions of reviewer #1 we will not show this figure in the final revised version, as it shows similar results as already presented in Lilienthal et al. (2018).

"4. Figure 3 and Page 7 Line 4: ". . . in the forcing locally amounts to +500% . . . and to . . . +1800% . . ..". The +500% and +1800% cannot be tell in Figure 3b and Figure 3h, although you have demonstrated the maximum and minimum values. You can add some red contours with the exact contour values in Figure 3 especially in Figure 3 (b, d, f, and h)."

Thank you for the suggestion, we will provide an updated version of the figure.

"5. You can simply illustrate the limit of the standard deviations. Only the interval of the standard deviations in Figures 1, 2, and 4 can not demonstrate the exact values."

In an updated version of the figures, we will also provide the maximum values of the standard deviations and amplitude differences (contour lines) in each figure panel.

"6. Page 7 Line 30: ". . .becomes instable for some months." The results of which months in Figure 6 are unstable?"

For the months June, July and August, Figure 6c and 6d do not show any data for an enhancement factor of 1.10. These are the simulations that became unstable. We will add a brief note in the revised version.

"Problems in figures and grammatical suggestions:

Figure 3: Figure description and titles of the Figure 3 are inconsistent. For example: 1. "July" in the figure description; "Jan and Apr" for the titles of Figure 3 2. "Zonal drag" due to GW-tide interactions in the figure description, "Zon. Acc." in the titles of Figure 3 3. a-h is not noted in Figure 3. There are similar problems in Figures 4, 5, 6."

We double checked all figures for the correct descriptions and included the a,b,c,... labels.

"Figures 6 and 7: 1. What's the difference of the vertical mean and global mean? The results of which latitude are demonstrated in Figures 6 and 7."

Figure 6 shows the horizontal mean (over all latitudes/longitudes) vertical mean (80-160km) amplitudes of the TDT. Figure 7 presents the horizontal and vertical mean of zonal mean wind/temperature differences, respectively.

"2. You can add some descriptions about the "factor of fgw" in the manuscript."

We will add a more precise description of that in the revised manuscript.

"Page 1 Line 4: "Scondar y sources" -> "Scondary sources" Page 1 Line 15: "the internal gravity waves (GW)" -> "the internal gravity waves (GWs)" Page 1 Line 16: "orography"->"geography" Page 2 Line 15: "be subject of" -> "be subject to"? Page 3 Line 18: "as described above" -> "as described in section 1" Page 4 Line 25 "rREF" -> "REF"; "maximums" -> "maxima" Page 5 Line 6: You presented the results at 109 km which can not represent the whole middle atmosphere. Thus, "in the middle atmosphere" is not exact in this sentence. Page 5 Line 6: "So the midlatitude" -> "For example, the midlatitude" Page 7 Line 25: "the respective slopes are given in the legend." -> "the respective slopes (s) are given in the legend.""

Thank you very much for all these technical corrections to improve the language. They will be addressed in the revised manuscript version.

---

## Author Response (AR1)

**Response to anonymous reviewer#1:**

Dear anonymous reviewer,

thank you very much for your valuable comments to help improve our manuscript. Please find below our detailed response to each of your concerns:

*The paper can be published, with the following revisions: The material reported in sections 1, 2, and 3.1 reprises findings reported in Lilienthal et al., 2018, and I saw little value in it. I recommend that section 3.1 be cut, or very briefly summarized. The paper should only report the new information appearing in section 3.2 and beyond, and Figures 3 and onward.*

> We agree that sections 1, 2 and 3.1 present some similarities with Lilienthal et al. (2018). However, in contrast to the present paper, there we presented simulations with one of the forcing mechanisms removed and therefore, two of three mechanisms were still included. The new aspect of Figs. 1 and 2 is to present the absolute contribution of each single forcing. We removed Fig. 1 from the manuscript and reduced Fig. 2, restricting ourselves to January conditions. The removed figures are now placed in the supplement along with the additional figures for April, July and October according to the updated Figs. 1-4. The text of the first sections, in particular section 3.1, was adjusted and shortened.

*I also recommend that it be edited for proper English grammar and usage.*

> We carefully went through the paper and corrected to our best knowledge, also considering the suggestions of reviewer #2 (please see tracked changes further down).

*Minor comments:*
*1. The text refers to panels a and b in Figure 4, but these labels that do not appear in the plots. Please recheck all Figures for this issue.*

> Thank you very much for the hint. We double checked all figures and included the labels in each of them.

**Response to anonymous reviewer#2:**

Dear anonymous reviewer,

thank you very much for your valuable comments to help improve our manuscript. Please find below our detailed response to each of your concerns:

*1. The model does not generate non-migrating tides in the current configuration as described in section 2. Please add "migrating" before "terdiurnal solar tide" in the title.*

> We adjusted the title, accordingly.

*2. "In order to reduce the time of computation for the simulations with enhanced forcing mechanisms, only the January runs are performed as an ensemble. The other months represent the conditions for the year 2000, only". Please discuss the potential difference or evaluate the influence of these two kinds of processing methods.*

> In the meantime, we can provide an update with ensemble means for the 5% enhanced simulations, too. This should be more consistent than using only one single year. The present Figs. 2-4 and related descriptions have been updated, accordingly (note that former Fig. 1 was removed following the suggestion of reviewer #1).
> Fig. 5 and 6 are still performed for the year 2000 and not based on ensemble simulations, but Fig. 1 of the manuscript includes a standard deviation with respect to the 11 ensemble members so that the reader may also get an idea of the year to year variability from this figure.

*3. Figure 1 and Page 5 Line 1: "This is due to the fact that MUAM tends to underestimate tides in general, which is frequently seen in other models, too".*
*Page 5 Line 14: There are differences in the seasonal variation of the TDT from different models. What's the reason behind these phenomena?*
*Besides, would you please demonstrate why you present the seasonal cycle of TDT at 109 km?*

> There is no general agreement on the cause of tidal underestimation in models, but it has to be connected with strong damping in the mesosphere region, for example due to gravity waves. The Lindzen-type parameterization used in MUAM tends to damp tides relatively strongly, while other parameterizations can lead to slightly stronger tides (e.g. the one after Hines or Yigit). Neverless, as we mainly focus on relative amplitude changes, our results are not significantly influenced by this underestimation.

> Please note that our statement about the different seasonal cycles (page 5 line 14 of the discussion manuscript) only refers to the pure nonlinear TDT, not to the total TDT (which could be compared to observations). The reason is most likely based on the different tidal forcing mechanisms, but also on different methods to determine the nonlinear TDT contribution. For example, Smith and Ortland (2001) used a model with explicit lower boundary forcing of the diurnal (DT) and semidiurnal tides (SDT), while all tides in MUAM are forced in-situ by absorption of solar radiation without lower boundary forcing. The latitudinal distribution of the tides by Smith and Ortland (2001) is therefore already prescribed to a certain degree and possible nonlinear interactions may appear at different latitudes and altitudes than in our model.

For Fig. 1, the altitude of 109km was chosen to be able to compare the REF results with satellite measurements which are not available further above. Below that altitude, MUAM produces relatively small amplitudes which might be connected with large uncertainties. Therefore, we think that 109km is the best choice as a compromise between large amplitudes and comparability. Note, however, that according to the suggestions of reviewer #1 we removed this figure in the revised version, as it shows similar results as already presented by Lilienthal et al. (2018). Instead, we added a supplement, where it can still be found.

*4. Figure 3 and Page 7 Line 4: ". . . in the forcing locally amounts to +500% . . . and to . . . +1800% . . ..". The +500% and +1800% cannot be tell in Figure 3b and Figure 3h, although you have demonstrated the maximum and minimum values. You can add some red contours with the exact contour values in Figure 3 especially in Figure 3 (b, d, f, and h).*

In order to improve the visibility of maximum values of Fig. 2 (former Fig. 3), we used a logarithmic scale instead of a linear one and included colors that are easier to distinguish. We believe that additional contour lines had been rather confusing in these plots.

*5. You can simply illustrate the limit of the standard deviations. Only the interval of the standard deviations in Figures 1, 2, and 4 can not demonstrate the exact values.*

We now provide the maximum values of the standard deviations and amplitude differences (contour lines) in the panels of Figs. 1 and 3.

*6. Page 7 Line 30: ". . .becomes instable for some months." The results of which months in Figure 6 are unstable?*

For the months June, July and August, Figure 6c and 6d do not show any data for an enhancement factor of 1.10. These are the simulations that became unstable. We added a brief note in the revised version (P8, L7-8).

*Problems in figures and grammatical suggestions:*

*Figure 3: Figure description and titles of the Figure 3 are inconsistent.*
*For example:*
*1. "July" in the figure description; "Jan and Apr" for the titles of Figure 3*
*2. "Zonal drag" due to GW-tide interactions in the figure description, "Zon. Acc." in the titles of Figure 3*
*3. a~h is not noted in Figure 3. There are similar problems in Figures 4, 5, 6.*

We double checked all figures for the correct descriptions and included the a,b,c,... labels.

*Figures 6 and 7:*
*1. What's the difference of the vertical mean and global mean? The results of which latitude are demonstrated in Figures 6 and 7.*

Figure 5 shows the horizontal mean (over all latitudes/longitudes) vertical mean (80-160km) amplitudes of the TDT. Figure 6 presents the horizontal and vertical mean of zonal mean wind/temperature differences, respectively. Therefore, "global mean" was here associated with "horizontal mean vertical mean". In the revised version, it has been substituted to avoid

confusion.

*2. You can add some descriptions about the "factor of fgw" in the manuscript.*

We removed the terms "fgw" and "fnlin" in Figs. 5 and 6 and used "terdiurnal GW forcing" and "terdiurnal nonlinear forcing", instead, to be more precise.

*Page 1 Line 4: "Scondar y sources" -> "Scondary sources"*
*Page 1 Line 15: "the internal gravity waves (GW)" -> "the internal gravity waves (GWs)"*
*Page 1 Line 16: "orography"->"geography"*
*Page 2 Line 15: "be subject of" -> "be subject to"?*
*Page 3 Line 18: "as described above" -> "as described in section 1"*
*Page 4 Line 25 "rREF" -> "REF"; "maximums" -> "maxima"*
*Page 5 Line 6: You presented the results at 109 km which can not represent the whole middle atmosphere. Thus, "in the middle atmosphere" is not exact in this sentence.*
*Page 5 Line 6: "So the midlatitude" -> "For example, the midlatitude"*
*Page 7 Line 25: "the respective slopes are given in the legend." -> "the respective slopes (s) are given in the legend."*

Thank you very much for all these technical corrections to improve the language. They have been addressed in the revised manuscript version.

**List of relevant changes**

**for the article "Nonlinear forcing mechanisms of the migrating terdiurnal solar tide and their impact on the zonal mean circulation" by F. Lilienthal and Ch. Jacobi**

Text:
- Section 3.1 has been shortened and adjusted because of removed figures.
- Grammatical and Spelling issues have been addressed.

Figures and Tables:
- a,b,c,... labels were added for all figure panels
- former Fig. 1 has been removed
- Fig. 1 (former Fig. 2) was shortened, now we only show January results
- Fig. 2: logarithmic scaling and new color scheme
- Figs. 2-4 are now based on ensemble simulations (GW5 and NL5)

Supplement:
- a supplement has been added to show also April, July and October conditions for Figs. 1-4
- the removed former Fig. 1 has also been added for zonal wind and temperature parameters in the supplement.

[revised manuscript text omitted]